# Detection of SARS-CoV-2 Based on Nucleic Acid Amplification Tests (NAATs) and Its Integration into Nanomedicine and Microfluidic Devices as Point-of-Care Testing (POCT)

**DOI:** 10.3390/ijms241210233

**Published:** 2023-06-16

**Authors:** Alexis Dorta-Gorrín, Jesús Navas-Méndez, Mónica Gozalo-Margüello, Laura Miralles, Lorena García-Hevia

**Affiliations:** 1Department of Molecular Biology, Faculty of Medicine, University of Cantabria (UC), 39011 Santander, Spain; alexis.dorta@alumnos.unican.es (A.D.-G.); navasj@unican.es (J.N.-M.); 2Instituto de Investigación Valdecilla (IDIVAL), 39011 Santander, Spain; monica.gozalo@scsalud.es; 3Environmental Genetics Department, Ecohydros S.L., 39600 Maliaño, Spain; 4Microbiology Service of University Hospital Marqués de Valdecilla (HUMV), 39008 Santander, Spain; 5CIBER de Enfermedades Infecciosas-CIBERINFEC (CB21/13/00068), Instituto de Salud Carlos III, 28029 Madrid, Spain; 6Department of Functional Biology, Area of Genetics, Faculty of Medicine, University of Oviedo, 33006 Oviedo, Spain

**Keywords:** COVID-19, diagnosis, clinical management, device integration, nanomedicine applications

## Abstract

The coronavirus SARS-CoV-2 has highlighted the criticality of an accurate and rapid diagnosis in order to contain the spread of the virus. Knowledge of the viral structure and its genome is essential for diagnosis development. The virus is still quickly evolving and the global scenario could easily change. Thus, a greater range of diagnostic options is essential to face this threat to public health. In response to the global demand, there has been a rapid advancement in the understanding of current diagnostic methods. In fact, innovative approaches have emerged, leveraging the benefits of nanomedicine and microfluidic technologies. Although this development has been incredibly fast, several key areas require further investigation and optimization, such as sample collection and preparation, assay optimization and sensitivity, cost effectiveness, scalability device miniaturization, and portability and integration with smartphones. Addressing these gaps in the knowledge and these technological challenges will contribute to the development of reliable, sensitive, and user-friendly NAAT-based POCTs for the diagnosis of SARS-CoV-2 and other infectious diseases, facilitating rapid and effective patient management. This review aims to provide an overview of current SARS-CoV-2 detection methods based on nucleic acid detection tests (NAATs). Additionally, it explores promising approaches that combine nanomedicine and microfluidic devices with high sensitivity and relatively fast ‘time to answer’ for integration into point-of-care testing (POCT).

## 1. Introduction

SARS-CoV-2 is a novel coronavirus that emerged in late 2019 and is responsible for causing the disease known as COVID-19. To date (11 February 2023), the coronavirus SARS-CoV-2 pandemic has resulted in 677,367,334 confirmed cases and 6 million deaths around the world. The United States, India, and France have been the most affected countries, with the USA reporting the highest number of deaths [1].

Knowledge of the SARS-CoV-2 genome and structure is essential for its diagnosis, therapeutic targets, pathophysiology, and genetic variation. SARS-CoV-2 is a betacoronavirus of the order *Nidovirales.* SARS-CoV-2 is the seventh coronavirus described as capable of infecting humans and the third capable of large-scale spread and pandemic disease, as did SARS-CoV in 2003 and MERS-CoV in 2012. The SARS-CoV-2 genome is a large, single positive-sense RNA strand of 26–32 kb, capped and polyadenylated, encoding 16 non-structural genes at the 5′ end, and 4 structural genes (S, M, N, and E) and 11 accessory proteins (ORF3 to ORF10) at the 3′ end [2]. This structure allows the viral genome to be translated as mRNA, which is directly recognized by the ribosomes of cells [3] (Figure 1).

Viral RNA polymerases exhibit a low fidelity rate, leading to the introduction of mutations during each replication cycle [4,5,6]. Alongside the rapid replication of the virus, its large population size and other mechanisms, such as viral recombination and reassortment, contribute to the high diversity observed in coronaviruses, aligning with the concept of quasispecies [7]. Consequently, this favors the emergence of variants of concern (VOCs). The World Health Organization (WHO) defines VOCs as variants that exhibit increased transmissibility, detrimental changes in COVID-19 epidemiology, increased virulence, altered clinical presentation, reduced effectiveness of public health interventions (e.g., social measures, diagnostics, therapeutics, vaccines), or any combination thereof [8].

Genomic sequencing and the deposit of genomes into databases such as Pango [9], Nexstrain [10], and the Global Initiative of Share All Influenza Data (GISAID) [11] have made it possible to monitor, characterize, and classify the epidemiological evolution of SARS-CoV-2 in different VOCs. The principal sublineages of Omicron currently circulating are shown in Table 1.

The Omicron sublines are now classified as VOIs (variants of interest) and VUMs (variants under monitoring). The differences between VOIs are defined as variants that show preliminary evidence, including genomic, epidemiological, or in vitro data, suggesting a potential impact on transmissibility, severity, or immunological escape, but with greater uncertainty. VUMs, on the other hand, are additional variants that may have similar characteristics to VOCs but lack strong evidence or proper evaluation.

Risk assessment and classification as a VOI or VUM by the WHO or ECDC may vary among the sublineages of Omicron [14]. The concern with the Omicron variant lies in its more than 30 mutations in the spike protein, a key surface protein of coronaviruses responsible for binding to the ACE2 receptor on the host cells [15,16]. These mutations contribute to increased infectivity, enhanced transmissibility, and potential immune evasion, which could reduce vaccine efficacy [17,18]. Certain sublineages of Omicron, such as BA.4 and BA.5, have shown a loss of diagnostic accuracy and potential false-negative results in silico in RT-qPCR assays [19,20]. Recombination phenomena between these sublineages are also commonly observed.

Therefore, accurate and updated nucleic acid amplification tests (NAATs), such as RT-qPCR, are necessary to monitor the evolution of the virus. Hence, the diagnosis and surveillance of SARS-CoV-2 variants are crucial for controlling the spread of the virus and adapting public health measures [21,22]. The synergies between NAATs and new technical advances in medicine, such as microfluidics and nanomedicine, will be reviewed here, offering an updated scenario of promising approaches and highlighting emerging trends in diagnostic methods, which will drive future advancements not only in the diagnosis of SARS-CoV-2, but also in identifying newly emerging or re-emerging pathogens.

## 2. Importance of Diagnosis and the Use of POCT

As an emerging pathogen, SARS-CoV-2 has demonstrated the importance of an early and accurate diagnosis, the stratification of disease severity, and clinical management. Symptoms of COVID-19 can be easily confused with other respiratory infections and infected individuals may be asymptomatic (without clinical signs but able to spread the disease). Early accurate diagnosis, the detection of suspected cases, with or without symptoms, and appropriate clinical management are therefore essential to break the transmission chain and implement social interventions where necessary.

According to the WHO, molecular detection methods for SARS-CoV-2 can be divided into three categories:-Detection of viral RNA: this involves nucleic acid amplification tests (NAATs), such as RT-qPCR.-Detection of viral antigens: such as immunodiagnostic techniques, including lateral flow assays (LFA).-Detection of viral antibodies: serological techniques, such as enzyme-linked immunosorbent assays (ELISAs) or chemiluminescent immunoassays (CLIAs).

Briefly, antigen tests, which have been widely available during the pandemic, utilize immunoassays to detect viral antigens. Positive results are indicated by colored bands on the test line, while a control line confirms the test’s accuracy [23]. Antigen tests generally have a lower sensitivity compared to NAATs, and their performance can vary among the different tests [24]. Moreover, the policies regarding the use of LFAs differ from country to country and have changed throughout the course of the pandemic. For instance, at the beginning of the pandemic, a confirmatory NAAT test was required after a positive LFA test, whereas nowadays an LFA test alone is considered sufficient.

The test defined as POCT, also called near patient or bedside testing, is one that can be performed in an outpatient setting, thereby shortening the clinical decision-making process for additional testing or therapy [25]. An ideal POCT test fulfils the REASSURED criteria of having REal-time connectivity, and being Affordable, Sensitive, Specific, User-friendly, Rapid and robust, Equipment-free, and Delivery to the end user [26]. The integration of the diagnosis into POCT devices has been possible thanks to the development of microfluidic technology and nanomedicine. Microfluidics is a field that deals with the behavior, manipulation, and control of fluids at the microscale level. It involves the design and fabrication of devices that manipulate tiny amounts of fluids, typically on a microliter or nanoliter scale, within microchannels or microstructures [27]. Nanomedicine, on the other hand, refers to the application of nanotechnology in medicine. It involves the use of nanoscale materials and devices for the diagnosis, treatment, and prevention of diseases [28].

The relation between NAATs and microfluidics or nanomedicine lies in their potential synergies and applications in the field of molecular diagnostics. Microfluidic systems can be used to miniaturize and automate NAATs, enabling the rapid and efficient analysis of nucleic acids with reduced reagent consumption. Microfluidic devices can integrate the various steps of NAATs, such as sample preparation, nucleic acid extraction, amplification, and detection, into a single chip or platform [29].

## 3. Nucleic Acid Amplification Tests (NAATs)

### 3.1. RT-qPCR Is the Gold Standard Method

Quantitative fluorescence-based reverse transcription-polymerase chain reaction (RT-qPCR) is widely acknowledged as the gold standard for SARS-CoV-2 detection and is extensively utilized in clinical settings due to its high sensitivity, specificity, and automation. The first step of sample preparation for RT-qPCR is the extraction of nucleic acids, specifically RNA for SARS-CoV-2, which can be time-consuming, labor-intensive, and prone to contamination. This is followed by the reverse transcription of the viral RNA into complementary DNA (cDNA). The third step is the qPCR amplification of the cDNA using specific primers. Both reverse transcription and amplification can be performed in a single tube. During each amplification cycle, the probe of the reaction, typically a TaqMan probe, hybridizes with the amplicon, and its hydrolysis emits fluorescence of different wavelengths. A real-time thermocycler monitors and records this fluorescence, allowing for the determination of the cycle threshold (Ct). Ct represents the number of cycles in which the fluorescence significantly exceeds the background, and enables the measurement of the exponential accumulation of amplicons. By comparing the Ct values of the controls and samples, the relative expression can be estimated [30,31]. The ability to multiplex the RT-qPCR reaction enhances its capacity to detect various targets of viral nucleic acid and improves the throughput of the assay. The automation of RNA extraction and RT-qPCR reduces time, manual labor, contamination risks, and handling errors. However, it requires expensive and bulky equipment.

In a diagnostic context, caution should be used when interpreting the Ct result. The Ct value does not represent infectious viral particles, nor the amount of viral RNA in the sample. A low Ct may be obtained when the viral load is high, and a high Ct may be obtained when the viral load is low, so the Ct must be interpreted in the clinical context of the patient. For that reason, RT-qPCR usually is used as a qualitative method (a yes or no answer), differing within different countries. Furthermore, a positive RT-qPCR test does not necessarily mean that the patient was infectious; the test may remain positive for 5 weeks after the onset of symptoms. There is variability between the RNA extraction method, inter- and intra-test, and heterogeneity in sample collection in the different commercialized RT-qPCR tests available; each laboratory must evaluate the Ct cut-offs and validate the viral load. This is an important limitation [32,33,34,35]. Other limitations of the test are the need for well-trained personnel to perform it, the cost, the large equipment, and the time required. Results can be delayed by days due to the transport of samples and time to perform the test, which could lead to new infections [36,37].

Based on the first sequences of SARS-CoV-2 deposited in the GISAID database in January 2020, the WHO published an RT-qPCR protocol for the detection of the E and RdRp genes. The primers for the assay were obtained from the National Reference Centre for Respiratory Viruses, Institute Pasteur, Paris, and Corman [38,39]. On the other hand, the US CDC developed different protocols considering different genes than the WHO. However, quality control problems were found and the protocol was re-analyzed. These issues, along with the emergency public health situation, caused a delay in diagnostic tests at the beginning of the pandemic. As a result, the Food and Drug Administration (FDA) changed its policy and allowed other laboratories outside the US CDC to perform and validate COVID-19 diagnostic tests. Subsequently, biotechnology companies around the world made efforts and competed to validate and commercialize their diagnostic tests. The CDC website provides a current and updated list of diagnostic tests, the majority of which are based on the FDA-approved RT-qPCR, classified by the entity, date of approval, type of test, and authorized settings (different laboratories or patient-care settings certified according to the Clinical Laboratory Improvements Amendments of 1988 (CLIA)). The list also includes information on the performance of the tests based on their suitability for specialized laboratories, fact sheets for authorized end users, healthcare providers, and patients, as well as manufacturers’ instructions. As for the POCTs that use RT-qPCR, the FDA has approved various tests, such as Xpert Xpress CoV-2 Plus and Xpert Xpress SARS-CoV-2/Flu/RSV tests from Cepheid GeneXpert, which include Influenza A/B and Respiratory Syncytial Virus [40], Accula SARS-CoV-2 test from Mesa Biotech, and a multiplex test for SARS-CoV-2 & Influenza A/B from Roche Molecular Systems, among others [41].

### 3.2. Digital Droplet Polymerase Chain Reaction (ddPCR)

The method that comes closest to absolute quantification is ddPCR (digital droplet polymerase chain reaction). In ddPCR, the sample is divided into droplets, each containing few or no copies of the target DNA. These droplets act as separate micro-reactions of the PCR and are then compared by measuring the fluorescence. The partitions follow a Poisson distribution, so the ratio of positive partitions to the total number defines the amount of target DNA in the sample. ddPCR is less affected by the presence of inhibitors than qPCR, although it depends on the specific inhibitor used. Additionally, ddPCR is more reproducible than qPCR and can be integrated into microfluidic technologies or devices to adapt to the desired application [42,43] (Figure 2).

### 3.3. Isothermal Amplification

Isothermal amplification encompasses all the nucleic acid amplification tests (NAATs) that can be performed at a constant temperature, eliminating the need for a thermocycler. Since 1990, various isothermal amplification methods have been developed as alternatives to PCR and have shown great potential in the field of biomedicine. These methods can be integrated with nanoparticles, microsystems, and bioanalysis, offering new possibilities [44].

There is a wide range of isothermal amplification methods, each utilizing different enzymes, numbers of primers, temperatures, and reaction times. Compared to RT-qPCR, isothermal amplification methods generally exhibit lower sensitivity, reproducibility, specificity, and robustness. However, researchers have documented several strategies to address and improve these limitations [45].

The most widely used isothermal technique is reverse transcription coupled to loop-mediated isothermal amplification (RT-LAMP). In RT-LAMP, multiple sets of primers (typically four to six) are designed to partially bind to the target sequence. This amplification process generates a new region with self-complementarity, forming a loop structure that can be recognized by additional primers (L-primers), resulting in a cascade of amplification. The results of RT-LAMP can be detected using various techniques, including end-point electrophoresis, turbidimetry, colorimetry, intercalating agent dyes, and real-time fluorescence. One of the attractive features of RT-LAMP is its versatility, as the reaction can be performed in a simple heat block or thermal bath, making it suitable for resource-limited settings. Several methods have been developed for the colorimetric detection of SARS-CoV-2 using RT-LAMP, which allows the reaction to be completed within 30 min at 65 °C, without the need for complex laboratory reagents [46] or through turbidimetry [47]. Although challenging, multiplexing the RT-LAMP reaction is possible and can be coupled with sequencing, enabling large-scale testing and the monitoring of variants of concern [48,49] (Figure 3).

Recombinase polymerase amplification (RPA) is another isothermal method of NAATs that is simpler (fewer enzymes required) and more sensitive (it can detect fewer copies) than LAMP. The reaction is shown in Figure 3. The process starts when the recombinase usvX binds to the primers and ATP to form a complex. This complex recognizes the DNA target in the sample. When it recognizes the complementary sequence, it produces a strand displacement that is stabilized by the SSB proteins. The complex is then disassembled and DNA polymerase polymerizes the strands at the 3′ end. This happens in a cycle of exponential amplification until the ATP is used up. The amplification time is often 20 min [50]. Although it is a powerful tool, it has several limitations that need to be overcome. It requires the full kinetic optimization of the reaction. It does not provide a Ct-like RT-qPCR, but a time threshold based on real time. The entire manual performance, e.g., the mixing step, affects the kinetics of the reaction, so it is recommended to automate and carefully control it. The false-positive amplification rate is higher than for other isothermal techniques. The dyes normally used in qPCR, such as SYBR Green or TaqMan probes, do not work in RPA reactions; the TaqMan polymerases digest the strands displaced by their 5′-3′ exonuclease activity [51,52].

The strength of RPA is that it could be coupled with other systems to improve sensitivity, reduce false-positive rates, and automate reactions. It could be coupled to a lateral flow assay (LFA) like other NAATs and provide visual readouts as a qualitative method, or be automated in microdevices or point-of-care biosensors. Some strategies will be reviewed in the next sections, but examples of the approaches that include RPA with CRISPR-based methods are SHERLOCK, with advances as presented by Song et al. (2023) [53], NanoPEIAs, etc. [54,55].

Other isothermal methods, such as transcription-mediated amplification (TMA), strand displacement amplification (SDA), and helicase-dependent amplification (HDA), have not been as widely used and reported for the detection of SARS-CoV-2. Although these isothermal methods have emerged in recent years as alternatives or competitors to LAMP and RPA, they have not been widely used; perhaps the accessibility of the kits and the stringency of the reagents favor a ‘niche’ application, and present difficulties in overcoming ‘proof-of-concept’ and becoming widely used, as has been suggested, for example, for HDA [56].

### 3.4. CRISPR-Based Methods

Clustered regularly interspaced short palindromic sequence repeats (CRISPR) has revolutionized molecular biology since its discovery, mechanism elucidation, and application [57,58,59]. Known in the scientific community as ‘genetic scissors’, it enables genome editing, with revolutionary applications and ethical challenges [60].

CRISPR-Cas systems can be classified into three classes, according to the type of endonucleases (Cas), based on their complexity: Class I, Class II, and Class III. Each class is subdivided into several types according to the structure and sequence of the Cas proteins. For diagnostic purposes, the type V (Cas12) and type VI (Cas13) proteins of class II have been adapted because of their simplicity.

The simplest, type V, requires only crRNA (RNA-guided CRISPR system) and Cas12a protein [61]. In short, an RNA-guide (crRNA) containing the target sequence is delivered to the CRISPR/Cas system. The CRISPR/Cas system, with the cRNA, scans the sequence and, if the target is present, specifically cuts the sequence at that location. The reaction can be monitored using fluorescent molecular reporters, probes or linked to enzymes [62]. In terms of diagnostics, several methods have been developed that combine isothermal amplification with CRISPR technology, such as SHERLOCK (Specific High-sensitivity Enzymatic Reporter unLOCKing) and DETECTR (DNA endonuclease-targeted CRISPR trans reporter) (Figure 4).

Both assays have high sensitivity, can detect very low levels of viral RNA in the order of 2 aM, are rapid, allow visual signaling after performance, and could avoid complex laboratory infrastructure, which is attractive for POCT [63]. Published protocols have used RPA and SHERLOCK to detect SARS-CoV-2 with results in less than an hour and a setup time of approximately 15 min. The detection method is coupled to a lateral flow device (LFD) to provide visual readouts and even to develop the entire process in a single tube, thus avoiding cross-contamination [64]. Interestingly, another approach called CRISPR-SPADE (CRISPR Single-Pot-Assay-Detecting-Emerging VOCs) combines RT-LAMP with CRISPR-Cas in one tube, allowing the detection of different VOCs [65].

An advantage, but also a limitation, of CRISPR-based assays is the possibility of avoiding RNA extraction, which is a time-consuming and costly step of RT-qPCR, but can in some cases inhibit the reaction due to the presence of DNA nucleases in the sample. For example, the method called CASSPIT (Cas13 Assisted Saliva-based & Smartphone Integration Testing) allows the detection of SARS-CoV-2 in saliva samples without RNA extraction, just heating, and using CRISPR/Cas13 and LFA showed a sensitivity of 97%, in agreement with RT-qPCR. Coupled with a smartphone, it allows quantification of the results and achieves the ease of use and connectivity that POCTs must have [66].

Another interesting approach as a POCT is STOP COVID (SHERLOCK Testing-in- One-Pot), which combines RT-LAMP with SHERLOCK in a one-tube reaction, without the need for RNA extraction and lateral-flow strip reading, offering high sensitivity, no cross-reactivity with SARS-CoV or MERS-CoV, and the results can be adapted to a cartridge to avoid contamination and a mobile device for quantification [67].

CRISPR-based assays can be integrated into a large-sample processing system, such as CARMEN (for Combinatorial Arrayed Reactions for Multiplexed Evaluation of Nucleic Acids), which can evaluate 4500 samples at a time. It consists of microdroplets of each CRISPR/Cas13 reaction emulsified in oil, and droplets of the sample, which are mixed by an electric field in a microarray well of the chip, generate a barcode of fluorescence when the cleavage reaction does or does not occur. The fluorescence is recorded and measured with fluorescence microscopy [68,69] (Figure 4).

CARMEN is a powerful tool that has shown high-specificity, -sensitivity, and -accuracy comparable to RT-qPCR and sequencing [69]. In terms of POCT, the main limitations are the cost, the need for qualified personnel, and well-equipped central laboratories to perform this test.

A promising technology related to SHERLOCK is INSPECTR (Internal Splint-Pairing Expression Cassette Translation Reaction), a DNA hybridization-based sensor that detects RNA or DNA single base-pair sensitivity coupled with a bioluminescent signal, co-founded by the SHERLOCK developers and the Harvard Wyss Institute [70,71,72,73].

### 3.5. Next-Generation Sequencing (NGS)

As a centralized technique, NGS is the best diagnostic tool for knowing which pathogen or pathogens are present in samples, what mutations they have, and key information about pathogenesis and phylogeny.

NGS is the next step up from the traditional Sanger sequencing method, based on labeled dideoxynucleotides (ddNTPs) incorporated into a branded extension of DNA, each of which emits a different fluorescence. NGS platforms, such as Illumina and IonTorrent, and also third-generation sequencing, such as Nanopore or PacBio, are based on the construction of a library of fragments from the genome of the sample with adapters (barcoding) and the use of labeled deoxynucleotides, pH change, or electrical mobility throughout a nanopore channel each time a dNTP is incorporated, to distinguish each nucleotide in parallel. The raw data is compiled by specific software, resulting in reads. The number of reads overlaps into contigs. The contigs need to be assembled and mapped to build the whole genome. Coverage is the percentage of the genome that is statistically correctly identified and assembled [74].

In the early stages of the COVID-19 outbreak, NGS provided important information about SARS-CoV-2. It allowed us to identify its relationship to bat-SL-CoVZC4, a bat coronavirus that has a protein spike more like the protein of SARS-CoV that caused the 2002 outbreak [75]. It also provides important information about the pathogenesis of viral RNA, such as the presence of the furin cleavage site in the spike [76]. The mutations along the genome that enhance viral infectivity have also been predicted by NGS, and key viral processes, such as viral binding to ACE2, have been described by using NGS [77]. Finally, the structure of the viral genome and the deposit of the sequences in databases, such as GISAID and Nexstrain, have allowed the development of other diagnostic methods, such as the first RT-qPCR and other NAATs [78,79].

Currently, in a clinical context, positive patients with a Ct of less than 30 are selected as candidates for sequencing in order to track the locally circulating variants of concern. The WHO has published a guide for NGS-based epidemiological surveillance of different VOCs to assist clinical settings in tracking circulating VOCs [80].

## 4. Microfluidics Integration and Nanomedicine Advances

The development of microfluidic technology and microfabrication processes has enabled the creation of nanoscale lab-on-chip devices based on various molecular techniques, including NAATs. Microfluidics allows the miniaturization, integration, and portability of complex laboratory reactions, reducing cost and time to answer [81,82]. The microfluidic platform is mainly driven by capillarity, pressure, centrifugal forces, electrokinetics, or acoustic waves. These characteristics allow low energy consumption, portability/wearability, lower cost of instrumentation, precision, and programmability [83]. Paper-based microfluidics driven by capillarity or gravity are called µPADs, and have many applications, not only in molecular diagnostics, but also in drug detection and environmental monitoring [84].

The potential and development of nanomedicine have been increasing in biomedical settings in recent years. Nanoscale materials possess inherent properties that are interesting and suitable for biological systems in terms of compatibility, manipulability, and functionality [85]. When combined with molecular methods, such as NAATs, nanomedicine helps overcome principal limitations. For instance, nanomaterials simplify sample collection and preparation, eliminating the need for time-consuming and laborious nucleic acid extraction in some cases. This combination with NAATs results in greater specificity and sensitivity, which are required for reliably detecting low viral loads and reducing the time to obtain results. Moreover, NAATs can assist in overcoming the drawback of poor signaling from a biosensor by amplifying it through a NAATs reaction [29,86].

However, these advantages are still being further studied along with other potentially important characteristics, such as device miniaturization. Miniaturization enables robust diagnostics without compromising sensitivity and specificity, making it feasible to deploy them in resource-limited settings. Cost-effectiveness and scalability pose challenges in nanomedicine, as the characteristics of nanoparticles, such as size, need to be individually assessed to ensure affordability and suitability for mass screening, thereby ensuring health accessibility [87,88]. This can add complexity to the technique, emphasizing the importance of knowledge and expertise in improving these aspects.

Moreover, the development of nanoparticles (NPs) has been widely used in biomedical settings, including SARS-CoV-2. Both dendrimers and polymersomes have been proposed as potential treatments or for the development of new vaccine formulations [89,90]. Even the current Pfizer/Biotech mRNA vaccines are based on liposomes [91]. Inorganic nanoparticles, including quantum dots, have been used to perform fluorescence immunochromatography combined with isothermal amplification and CRISPR-based assays to detect SARS-CoV-2, with high sensitivity and a result time of 40 min [92]. Other approaches using silver and gold NPs will be reviewed below. Microfluidic mixing-based fabrication methods offer better control for achieving the desired size, morphology, shape, size distribution, and surface properties of the synthesized NPs [93].

Nanoplasmonics is an optical phenomenon in which nanoscale light interacts with a metal surface, causing the conversion of free photons into localized oscillatory-density charges on the metal’s surface (plasmonic surface). The metals commonly used for this phenomenon are gold and silver, although aluminum and copper can also support plasmonic resonance. In a colloid solution, plasmonic-charged nanoparticles act as biosensors. When a biological target of interest, such as viral RNA, binds to the surface of the nanoparticles, it induces a change in the refractive index. This change is reflected in the electromagnetic field and can be measured by tracking the resonant wavelength in the spectrum of scattered or transmitted light [94,95]. An example of the use of nanoplasmonics in detecting SARS-CoV-2 is the study conducted by Huang et al., which employed a nanoplasmonic-sensor chip functionalized with captured antibodies and gold nanoparticles functionalized with the ACE2 receptor. This approach enabled the detection of a SARS-CoV-2 pseudo-virus in the range of 0–1.6 × 10^10^ viral particles/mL within 15 min, demonstrating high specificity compared to SARS and MERS-CoV [96] (Figure 5).

Another technique is Raman spectroscopy, which is based on measuring the radiation emitted when a solution is excited by infrared light. When the solution is excited, the electrons within it are moved to higher energy levels, and their relaxation produces inelastic and elastic wavelengths. The inelastic wavelengths are recorded in the spectroscopy data and provide a fingerprint spectrum of the molecules and molecular bonds present in the solution [97]. Raman spectroscopy has been used to detect SARS-CoV-2 in human blood serum, with distinct spectra observed among healthy, infected, and suspected cases, showing high sensitivity in distinguishing among different groups [98] (Figure 5).

In summary, by combining the advantages of microfluidics and the capabilities of nanoparticles, several point-of-care testing (POCT) devices have been developed. The well-known µPADs are widely used for antigen tests. Paper-based antibody tests, such as serological IgG detection of SARS-CoV-2, have also been developed, improving the sensitivity of impedance electrochemical biosensors based on zinc nanowires and overcoming their limitations [99].

One example of integration is the use of a multiplexed CRISPR-based assay with RT-RPA as a µPAD, for the simultaneous detection of the N and S genes of SARS-CoV-2, with the RNAse P human gene serving as a control. A programmable sucrose valve separates the two reactions, allowing RPA amplicons to move into separate paper chambers where CRISPR cleavage occurs. Detection is based on fluorescence, and the entire process takes just 1 h with a sensitivity of 102 copies of the viral genome [100].

Another example of integrating different techniques is the Dµchip. The Dµchip integrates RT-LAMP and CRISPR/Cas12 in a chip with screw valves, where the reagents are mixed in the bottom chamber with the top microchip. This integration allows for the simultaneous detection of SARS-CoV-2 and different influenza viruses in a portable device that measures fluorescence. The Dµchip demonstrates high sensitivity and specificity [101]. RT-LAMP has also been integrated into a 3D cartridge chip, along with a smartphone as a reader, enabling RNA extraction-free detection within 40 min [102].

RT-RPA has also been combined with lateral flow in a highly sensitive assay which is able to detect as little as one copy of each variant, with no cross-reactions with other respiratory viruses, and has a performance time of 25 min, with visual readouts [53].

The combination of plasmonic resonance with gold nanoislands (gold AuNIs) functionalized with complementary DNA receptors can provide for the sensitive and specific detection of RNA viruses using acid nucleic hybridization. The plasmonic photothermal energy can improve the discrimination of different gene sequences in situ, allowing the detection of 0.22 pM of the precise targets in a raw sample [103]. Other developments have combined the AuNPs in a particle bioinspired in a virus that interacts between them and the spike protein of SARS-CoV-2, creating a plasmonic gap. These plasmonic gaps cause an extinction peak near infrared light and could be measured in a micro-optoelectronic chip and coupled to a smartphone, with a detection limit of 1.4 × 10^1^ pfu/mL [104].

Biosensors, in the form of nanoparticles of glass slides functionalized with specific SARS-CoV-2 probes immobilized on their surface and integrated into a microfluidic platform, are able to detect RNA/DNA duplexes with SYBR green from raw saliva samples, with a detection limit of 10 aM [105]. Another approach is the use of a DNA walker that binds to a silver-coated glass slide, with an enzymatic reaction of exonuclease that allows the release of DNA sequences if the target is present, by hybridization and cleavage of the exonuclease. The fluorescence emitted correlates with the amount of target present, and can be quantified using a smartphone [106].

Biosensors can enhance Raman spectroscopy, called SERS (surface-enhanced Raman spectroscopy) biosensors, based on a gold nanoparticle layer with antibodies to the spike protein, and a Raman reporter-labeled silver nanoparticle with an ultra-high sensitivity of 6.07 fg mL^−1^ in untreated saliva [107]. Another example is NanoPEIA (nanoplasmonic enhanced isothermal amplification), a nanoplasmonic chip array functionalized of gold with thiolated primers mixed with lyophilized reagents of RPA and the synthesis of DNA over the surface of the chip. A fluorescence resonance energy transfer (FRET) probe is a real-time reporter of the reaction. This reaction can be coupled to a high-throughput detector, visual detection, or POCT diagnostic platforms with 100% sensitivity to detect gene N and orfab1, and has a limit of detection of 28.5 and 23.3 copies per milliliter, respectively [108]. Using centrifugal microfluidics, RT-RAA (recombinase-aided amplification) has been integrated with CRISPR to detect the SARS-CoV-2 gene E ultra-sensitively, with an LoD of one copy per microlitre and 30 min of reaction time [109].

The combination of electrochemical microfluidics and nanoparticles created eCovsens, which immobilize SARS-CoV-2 monoclonal antibodies on a screen-printed carbon electrode (SPCE) and detects the antigen of the protein spike S1. The antigen–antibody binding generates an electrical charge that is measured in a few seconds, with a detection limit of 10 fM [110]. Other electrochemical approaches combine isochatophoresis (ITP), RT-LAMP, and CRISPR. Nucleic acid extraction using ITP and RT-LAMP amplifies the E and N genes and the molecular cleavage of CRISPR. Then, an electric field chip controls the reagents of CRISPR and a produces a fluorescence readout, with a performance time of 40 min and a limit of detection of 10 copies per microlitre [111]. Other recent integrations of CRISPR assays with electrochemical sensors have been made using aptamers with a high affinity to the S1 domain of the spike protein in complex biological fluids as the raw samples. CRISPR cleavage is detected via the binding aptamer S1, and measured using electrochemical impedance spectroscopy and differential pulse voltammetry, with an ultra-high sensitivity of 1.5 pg/mL. It has been tested in several variants of interest as a promising POCT [112,113].

Capacitive biosensors that react to metal by proximity above a certain capacitance were developed to detect SARS-CoV-2. A gold interdigitated electrode with antibodies immobilized on its surface was used to detect the spike protein by measuring capacitance changes. The biosensor showed high selectivity for Zika and Dengue viruses with no cross-reactivity [114].

The Wyss Institute developed a wearable mask combining RT-RPA and SHERLOCK in µPADs membranes separated by polyvinyl alcohol and an LFA strip for visual readout. It is able to detect SARS-CoV-2 through the respiratory aerosols by pressing a button, in a wearable format with high sensitivity and specificity, with a 90 min performance time [115]. Table 2 summarizes the different approaches presented in this review and compares them to commercially approved POCTs, such as Cepheid Xpress or Abbott ID Now.

## 5. Discussion

The pandemic caused by SARS-CoV-2 has accelerated scientific knowledge in the field of diagnosis due to the urgency and severity of the disease. This increase has been possible thanks to scientific data sharing and multiple efforts to understand the biology of the virus: its genome and viral structure, how it enters its host’s cells, its process of viral replication, and its phylogeny to understand the origin [2,3,118,119].

The strategy of test-trace-isolate is fundamental to contain the spread of a virus, complemented with other social measures [120,121]. In the first stage of the pandemic, RT-qPCR and NGS by themselves did not provide the fast response required to correctly isolate asymptomatic infected individuals, due to delays from transport sampling, reagents scarcity, the complexity of performing the test at centralized laboratories, and the need for well-prepared professionals [36,37]. In this situation, governments and health authorities created special funding calls to accelerate research on new methods of diagnosis and special POCT systems to control the widespread infection and mitigate its effects. Thus, all the diagnosis methods suffered from the importance placed on quick advancement, which was supported by biotechnology companies, and behind that was accuracy, speed, sensitivity, and specificity. Even RT-qPCR methods improved the time of performance, by lyophilizing the reagents and bringing to the market different portable thermocyclers to adapt the test as a POCT [122].

After extensively reviewing the various promising approaches that combine multiple molecular methods with ultra-high sensitivity, such as RT-RPA-LF, biosensor glass slides, centrifugal microfluidics, and quantum dots, we noted that some of these approaches relied on synthetic oligonucleotides or infected saline buffers instead of clinical samples for detection. However, it is important to highlight that clinical validation is the subsequent critical step following analytical validation, essential for a diagnostic technology to achieve the necessary level of development for its widespread acceptance and commercialization. This process presents real challenges, including the rigorous evaluation of infected and healthy patients, meticulous documentation and traceability, adherence to stringent parameters, compliance with country-specific regulations, and necessitates a multidisciplinary approach involving nanomedicine experts, engineers, molecular biologists, and clinicians.

Isothermal amplification and the use of nanomaterials and microfluidics have improved the detection of nucleic acids, providing high sensitivity, lower cost, and faster results [56,84,95,96]. Another important issue in diagnosis is the type of sample. The ideal sample is the least invasive that can be collected by the patient, preferably saliva rather than swabs or blood. Sample preparation must be minimal to avoid RNA extraction, which is a time-consuming step and affects sensitivity, depending on the extraction method [122]. Many of the approaches we have reviewed have taken this into account and can be performed rapidly, thanks to isothermal amplification kinetics and the tolerance of inhibitors, or the specificity of functionalized nanoparticles, such as glass slides or aptamers [53,66,104,105,108,109,112].

Accurate diagnosis is critical, and approaches have been focused on improving sensitivity and specificity. This allows us to avoid wrong diagnoses, as well as false negatives or false positives which, as we have explained, have undesirable consequences that affect clinical management. To minimize the occurrence of false negatives, several authors and commercial products have performed multiplexing of the targets in the test (different genes). We have reviewed some multiplexed approaches capable of detecting multiple targets of SARS-CoV-2, multiple respiratory viruses simultaneously, or different variants of concern within SARS-CoV-2 [65,66,67,98,101,103,108,111]. Multiplexed reactions within isothermal amplification can be a limitation, as it is possible that more unspecific amplification has occurred, as well as the competition between primers for each target, etc. CRISPR/Cas cleavage can help to increase specificity and minimize background noise.

The era of digital medicine is growing, probably accelerated by the SARS-CoV-2 outbreak and coupled with the massive development of the internet and smartphone accessibility. These technological developments are revolutionizing all fields, including health. Efforts to integrate microfluidic chips and transmit signals via smartphones are therefore evident, in line with the real connectivity promoted by the WHO for POCT devices. It allows for simplicity, lower cost, less energy, non-contamination, and direct detection. In addition, results can be obtained simultaneously by the end user and the clinician. It should not be forgotten that a diagnostic result alone is not enough; test results should always be evaluated in the clinical context. A few years ago in China, there was a trend towards home self-testing of biomarkers, which could be bought online or even in vending machines, and which promoted the accessibility and ownership of health by the end user. It also decentralized the healthcare system and could be useful for triage. However, it should not replace the work of doctors and healthcare professionals, who should be the final arbiters of clinical management. These tests should be complementary to POCT devices, and smartphone/artificial intelligence (AI) integration can help with epidemiological surveillance and the correct interpretation of results. AI is particularly relevant to approaches that integrate Raman spectroscopy or biosensors, due to their complexity. For example, AI can help to discriminate between the results obtained and to interpret the results for healthcare workers who are not specialized in the techniques. Even for qualitative methods, AI can help to recognize whether or not there is a measurable change in color, turbidity, or flocculation, simply because the interpretation of results by the ‘human eye’ can sometimes be confusing. AI is not subject to the subjective factors humans are and can dismiss inconclusive results and avoid repeating tests. Furthermore, integration with smartphones and AI could ensure that health accessibility is widely attained. The impact of the pandemic has been worse in low- and middle-resource communities, where health systems are as fragile as the economies of their inhabitants, with indigenous and small populations being the most affected [123]. Thus, in this scenario, POCT devices are more convenient due to the high demand; the microfluidic integration can reduce the energy demand, thus lowering the cost and eliminating the need for well-trained professionals to perform the tests. Linking to health centers can help in this scenario by tracking patients, controlling proliferation, and avoiding, for example, long journeys for residents to health centers or hospitals when this is not necessary. Finally, epidemiological surveillance of SARS-CoV-2 is still needed, and people living with long or persistent COVID should not be forgotten. The virus, as a quasispecies, may eventually mutate and VOC emergence is often likely. The prevalence of omicron lineages and the percentage of cases suggest that the virus may live with us for a long time with seasonal patterns. It is therefore necessary to trace variants and isolate cases as much as possible. The situation of long COVID is still under study, an understanding of its complexity and identifying its symptoms is difficult due to its heterogeneity [124]. However, the people who suffer from long COVID must receive an appropriate response from the health system and the scientific community. The development of real-connected POCT devices and diagnostic knowledge will help these people to communicate with the health system, helping to dismiss the impact of the disease in their lives, and aid in their recovery.

## 6. Conclusions

Diagnostic methods have evolved rapidly due to the pandemic, and have followed advances in nanomedicine and microfluidics, which allow for direct detection with lower costs, less energy, and no contamination. We have reviewed some promising diagnostic tools based on genetic-amplification detection tests, that could potentially become POCT tests combining new advances in the use of nanomaterials and digital medicine. POCTs had an opportunity for exponential grow during the pandemic as a qualitative method to rapidly test-trace-isolate. However, the development of any POCT must be clinically validated and this represents a real challenge, as it never replaces the role of the physician in evaluating the result of the test in its clinical context.

Nowadays, new genetic techniques, such as isothermal amplification, CRISPR-based methods, and NGS, have opened challenging opportunities to implement POCTs using nanomaterials and microfluidics towards a promising scenario where artificial intelligence (AI) integration can help with the correct interpretation of results and epidemiological surveillance. It is too soon to conclude which could be the most promising technique for developing the best POCT. However, after extensively reviewing the genetic detection approaches, it seems clear that a combination of multiple molecular methods allows for the performance of tests with ultra-high sensitivity and fast times. So, the future is undoubtedly linked to a multidisciplinary approach to bring together the advances of each field and each technique. Furthermore, technological developments have enabled the full integration of POCTs with microchip devices, offering high sensitivity and connectivity to smartphones. This integration enhances health accessibility, reduces costs, and facilitates a seamless integration into the healthcare system.

## Figures and Tables

**Figure 1 ijms-24-10233-f001:**
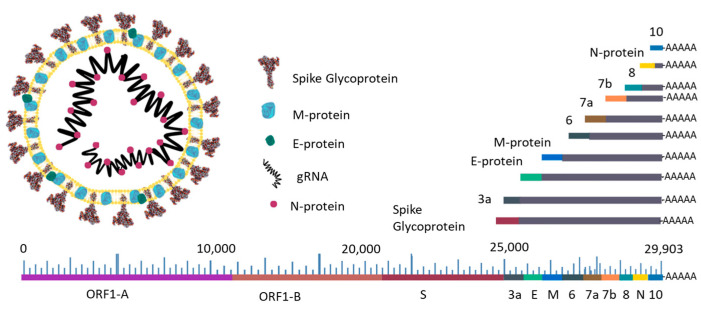
Schematic representation of the SARS-CoV-2 virion structure and genome. The subgenomic RNA transcripts of the structural genes and accessory proteins are represented by grey lines. Adapted from [3].

**Figure 2 ijms-24-10233-f002:**
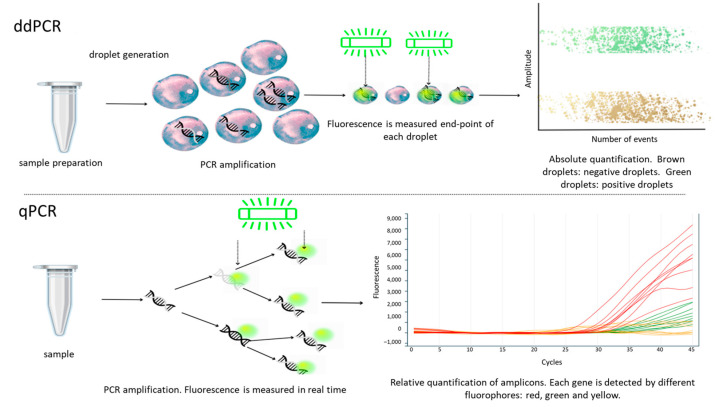
A schematic comparison of ddPCR and qPCR illustrates their differences in operation. In ddPCR, droplets are formed, each containing varying amounts of nucleic acid, enabling individual micro-PCR reactions within each droplet. The results are then analyzed based on Poisson’s distribution for accurate quantification. On the other hand, qPCR involves real-time amplification, where the fluorescent probe hybridizes and is continuously measured during each cycle. This results in an exponential graph if the result is positive, indicating relative quantification. The figure has been adapted from Kokkoris et al. [30].

**Figure 3 ijms-24-10233-f003:**
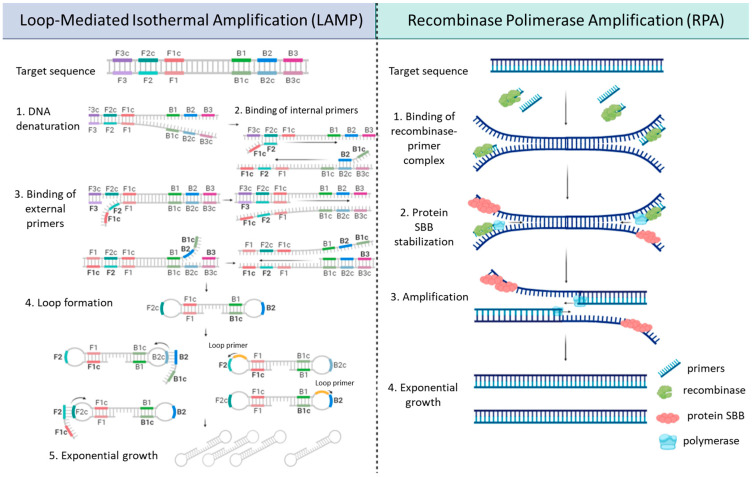
Schematic representation of LAMP and RPA. In LAMP, 4–6 primers are used to generate cohesive ends, allowing the formation of a loop structure and enabling exponential amplification. On the other hand, RPA employs two enzymes: a recombinase, which forms a complex with the primers and displaces the DNA strand, and a polymerase, which carries out the amplification process. Single-stranded DNA-binding (SSB) proteins are involved in stabilizing the single strand during the displacement step.

**Figure 4 ijms-24-10233-f004:**
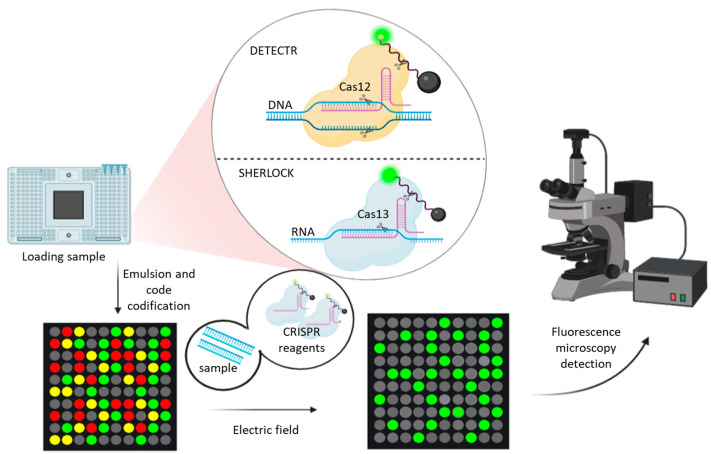
The DETECTR and SHERLOCK methods are schematically represented, with the main difference being the CRISPR/Cas system used. DETECTR utilizes CRISPR/Cas12 and targets double-stranded DNA, while SHERLOCK employs CRISPR/Cas13 and targets RNA. In both cases, when cleavage occurs, a fluorescent marker collaterally reports the cleavage and emits a signal. CRISPR/Cas systems can be integrated into microfluidic devices, such as the Combinatorial Arrayed Reactions for Multiplexed Evaluation of Nucleics Acids (CARMEN) method. Although theoretically possible with both CRISPR systems, CARMEN has been developed specifically for SHERLOCK assays. The sample is emulsified and barcoded to distinguish different targets or samples. An electric field combines each droplet with the necessary reagents for SHERLOCK. The presence or absence of a fluorescence signal allows for the quantification of positive samples for the target and enables multiplexed screening of several pathogens or targets of interest simultaneously.

**Figure 5 ijms-24-10233-f005:**
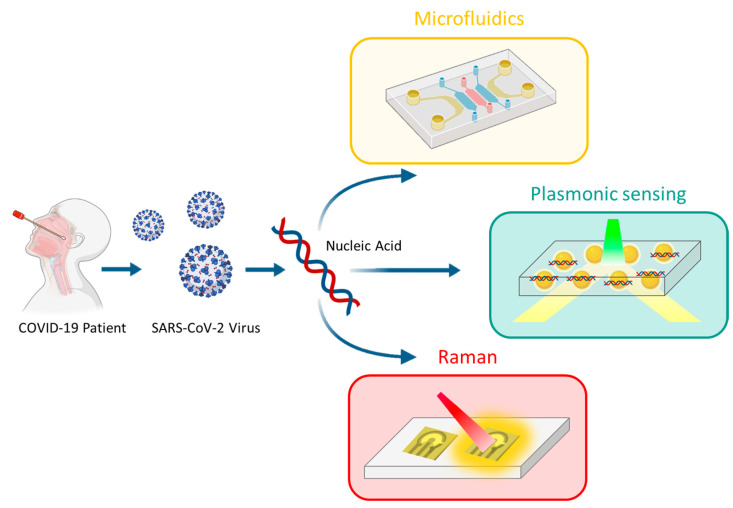
Schematic representation of the main nanomedicine-based molecular diagnostic systems for SARS-CoV-2 currently in use: microfluidics, plasmonic sensing, and Raman spectroscopy.

**Table 1 ijms-24-10233-t001:** The main variants of interest (VOIs) and variants under monitoring (VUMs) currently circulating are classified by Pango and Nextrain, and their designation and risk assessment data are available. These variants are characterized by genetic features, including the process of recombination that led to their emergence, and specific mutations in the spike protein. It is important to note that Lineage XBB does not include the sublineages mentioned as VOI and VUM (Source: WHO) [12,13].

VOIs			
Pango Lineage	Nexstrain Clade	Genetic Features	Date of Designation and Risk Assessment
XBB.1.5	23A	Recombinant of BA.2.10.1 and BA.2.75 sublineages, namely BJ1 and BM.1.1.1, with a breakpoint in S1 XBB.1 + S:F486P (similar spike genetic profile as XBB.1.9.1)	24 February 2023
XBB.1.16	23B	Recombinant of BA.2.10.1 and BA.2.75 sublineages, i.e., BJ1 and BM.1.1.1 XBB.1 + S:E180V, S:K478R and S:F486P	17 April 2023
**VUMs**			
BA.2.75	22D	BA.2 + S:K147E, S:W152R, S:F157L, S:I210V, S:G257S, S:D339H, S:G446S, S:N460K, S:Q493R reversion	6 July 2022
CH.1.1	22D	BA.2.75 + S:L452R, S:F486S	8 February 2023
BQ.1	22E	BA.5 + S:R346T, S:K444T, S:N460K	21 September 2022
XBB*	22F	BA.2 + S:V83A, S:Y144-, S:H146Q, S:Q183E, S:V213E, S:G252V, S:G339H, S:R346T, S:L368I, S:V445P, S:G446S, S:N460K, S:F486S, S:F490S	12 October 2022
XBB.1.9.1	Not assigned	Recombinant of BA.2.10.1 and BA.2.75 sublineages, i.e., BJ1 and BM.1.1.1 XBB.1 + S:F486P (similar spike genetic profile as XBB.1.5)	30 March 2023
XBB.1.9.2	Not assigned	Recombinant of BA.2.10.1 and BA.2.75 sublineages, i.e., BJ1 and BM.1.1.1, XBB.1 + S:F486P, S:Q613H	26 April 2023

**Table 2 ijms-24-10233-t002:** Comparative table of the different methods for the detection of SARS-CoV-2, in terms of the type of test, target, detection limit, time-to-result, and need for the pre-treatment of samples. * Some assays utilize lateral flow as a visual reading method, which can be considered part of the microfluidics group due to this characteristic.

Type of Test	Name/Manufacturer	Target	Limit of Detection	Time-to-Result	Extraction of RNA/Pretreatment	Reference
RT-qPCR						
RT-qPCR	Cepheid Xpress^®^ GenXpert	gene N, gene E	0.005 and 0.02 pfu/mL, respectively	45 min	Sample is mixed and transferred to the cartridge and loaded onto the system	[116]
**Isothermal amplification**						
RT-LAMP	Abbott ID Now™	gene RdRp	125 GE/mL with variations in different studies	≤13 min	Sample is transferred to the cartridge to the test base, initiating target amplification	[117]
RT-RPA/LF *	RT-RPA/LF	gene N	1 copy/µL	25 min	Without RNA (extraction infected samples with RNA)	[53]
**CRISPR-Cas based systems**						
CRISPR-Cas13/LF *	CASSPIT	genes S and N	~100 copies	≤1 h	Without RNA extraction (untreated samples)	[66]
**Isothermal amplification with CRISPR-Cas**						
RT-LAMP or RT-RPA/CRISPR-Cas13 *	SHERLOCK	DNA/RNA	2 aM	50 min	Without RNA extraction (heating samples)	[64]
RT-LAMP/CRISPR-Cas12b	CRISPR-SPADE	gene N of each VOC	15 copies/µL α 25 copies/µL β 50 copies/µL γ 12 copies/µL δ	30 min	Without RNA extraction (in vitro RNA transcripts chemically synthetized)	[65]
RT-LAMP/CRISPR-Cas13 *	STOP-COVID	gene N	100 copies of the viral genome	40–70 min depending on LF or fluorescence, respectively	RNA extraction using magnetic beads	[67]
**NAATs-Microfluidics**						
RT-RPA/CRISPR-Cas12a	µPAD CRISPR	genes S and N	100 copies of the viral genome	1 h	RNA extraction of 15 min	[100]
RT-LAMP/CRISPR-Cas	Dµchip	SARS-CoV-2, influenza A H1N1, H3N2 e influenza B RNA	10 copies	55 min	RNA extraction separated from the chip	[101]
RT-LAMP/cartridge and smartphone	RT-LAMP 3D cartridge	gene N	50 copies RNA (VTM), 5 × 10^4^ copies in nasal solution	≤40 min	Without RNA extraction (lysis 1 min using heat, 95 ºC)	[102]
RT-RAA/CRISPR-Cas	Centrifugal microfluidics	gene E	1 copy/µL	30 min	RNA extraction separated from the chip	[109]
Isochatophoresis-RT-LAMP/CRISPR-Cas	ITP/-RT-LAMP	gene E, gene N and human RNAse P	10 copies/µL	35 min	Without RNA extraction (untreated samples)	[111]
**Nanomedicine**						
gold nanoparticle layer with antibodies	SERS-biosensor	protein S	6.07 fg per mL	Not specified	Without RNA extraction (untreated samples)	[107]
AuNPs/plasmonic sensor and smartphone	Nanoplasmonic sensor	protein S	370 viral particles/mL	15 min	Without RNA extraction (SARS-CoV-2 pseudovirus)	[96]
Aptamers/CRISPR-Cas12a and potentiostat	Aptamers	S1 domain Spike	1.5 pg/mL	30 min	Without RNA extraction (untreated samples)	[112]
**Nanomedicine-microfluidics**						
Hybridization DNA walker with a functionalized glass slide	DNA walker/glass slide	two parts of RdRp gene	1.19 pM	30 min	RNA extraction with commercial kit	[106]
Hybridization with a functionalized glass slide	Biosensor glass slide	RNA/DNA	10 aM	15 min	Fast RNA extraction automated	[105]
RT-RPA in gold-layer functionalized chip	NanoPEIA	gene E and orfab1	28.5 and 23.3 copies per mL	6 min in Ct ≤ 25	Samples lysed 95 °C 5 min	[108]
FTO electrode functionalized with antibodies	eCovSens	antigen protein spike	90 fM	10–30s	Without RNA extraction (buffer samples and spiked saliva samples)	[110]
capacitive biosensor	Capacitive biosensor	protein Spike	≈760 pg/mL–76 ng/mL	15–20 min	Without RNA extraction (spiked sample in phosphate-buffered saline)	[113,114]
SHERLOCK in a face mask-integrated sensor	FDCF Wearable face mask	gene S	500 copies/17 aM	~1.5 h	Viral lysis with lyophilized compounds	[115]

## Data Availability

Not applicable.

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
