# Peer review of "Detection of SARS-CoV-2 Based on Nucleic Acid Amplification Tests (NAATs) and Its Integration into Nanomedicine and Microfluidic Devices as Point-of-Care Testing (POCT)"

_ijms, 2023, doi:10.3390/ijms241210233_

Round 1
Reviewer 1 Report
The authors describe NAATs for the detection of SARS-CoV-2. This work is not a significant contribution to the field, since there are already a lot of review available on this topic. Morever, extensive editing of the language is required. In more detail:
1. It is not clear what is innovative/new about this review in comparison with other reviews that are already published.
2. The text is very hard to read, due to missing structure (indicate more clearly what is discussed where, don't jump from subject to subject in one sentence/paragraph), long sentences, missing articles (mostly 'the'), using wrong forms (e.g. noun instead of the verb), wrong word order, mixing up singular and plural, or 'o' in the text. This is the case, among others, for the following sentences: R31-34, R41-46, R56-58, R69-70, R74-76, R96, R111-113, R117-118, R122, R125-136, R156, R178, R186-189, R198, R203-204, R230-231, R308-309, R332-334, R342-346, R348-351, R354-357, section 3.5 as a whole, R404-405, R412-413, R424-426, R427-431, R435, R458-462, ... (not further checked).
3. The first chapter/section has the wrong lay-out (and is different from the rest of the manuscript).
4. References are missing in the first part of the pagragraph starting at R35.
5. Table 1: Put the caption above the table. Why is the last row coloured? What does the * mean at Omicron?
6. The E in REASSURED is not indicated where it comes from.
7. R145: What about the position of the FDA?
8. Section 2: Why are only LFAs discussed here? Indiciate clearly the boudnaries of this review; what will be and what will not be discussed?
9. R179-180 and section 3.1: Give more details on the complete analysis process. Nucleic acid analysis is not only amplification, but also involves quite some sample work-up steps. Those steps usually involve quite some manual labour. So, also emphasize a bit more on what is meant with automatization.
10. R194: Emphasize more on the statement that the Ct-value might not be related to the viral load.
11. A ')' is missing somewhere around R220.
12. The Xpert Xpress is not the only POCT with authorization from the FDA.
13. There are no 'the' inhibitors (R234)
14. R257: this should be reverse instead of retro
15. LAMP can also be carried out with four primers.
16. What is meant with 'half complementarity' (R259)?
17. R266: Although multiplexing is possible with LAMP, it is very challenging!
18. From page 9 the lay-out is again incorrect.
19. There is no figure 5 about RPA. Performance is not expressed in amount of minutes. What is meant with scanning the DNA? Explain better statements such as 'simpler and more sensitive'. The use of lateral flow is not restricted to RPA only. Which SARS-CoV-2 test uses RPA? Please emphasize more on these things.
20. R311 mentiones type I, II and III and then only something is said about a type V, which is very confusing.
21. Lateral flow assay (LFA) is the correct term (R335).
22. MPS or NGS are more commonly used terms than HTS (section 3.5).
23. Figure 2: What does minimum pretreatment enhold? Give more details. Time-to-result is a more common term. Be consistent with capitals. Not necessary.
24. R547: I guess HIV is meant instead of VIH
25. The conclusion is missing an actual conclusion. What is the most promising technique?
The quality of the English is so poor that the article cannot be read properly. My advice it to check it by a native English speaker.
Author Response
Comments and Suggestions for Authors
REFEREE #1
The authors describe NAATs for the detection of SARS-CoV-2. This work is not a significant contribution to the field, since there are already a lot of review available on this topic. Morever, extensive editing of the language is required. In more detail:
- It is not clear what is innovative/new about this review in comparison with other reviews that are already published.
Reply to referee. Thank you for the comment. Our objective in undertaking this review is to provide a comprehensive overview of the current diagnostic methods for detecting SARS-CoV-2, with a particular emphasis on their clinical applications and integration with nanomedicine and microfluidics. Our team consists of professionals from diverse fields, including physicians, molecular biologists, and nanomedicine experts. Through our collective expertise, we have identified a knowledge gap regarding the advancements in these disciplines. Consequently, our aim is to develop a review that is accessible, not overly technical, and can be useful not only to us but also to others seeking to identify potential research areas and the convergence of these aforementioned disciplines. Additionally, we intended to highlight emerging trends in diagnostic methods that we believe will shape future advancements, benefiting the diagnosis of not only SARS-CoV-2 but also new emerging or re-emerging pathogens.
- The text is very hard to read, due to missing structure (indicate more clearly what is discussed where, don't jump from subject to subject in one sentence/paragraph), long sentences, missing articles (mostly 'the'), using wrong forms (e.g. noun instead of the verb), wrong word order, mixing up singular and plural, or 'o' in the text. This is the case, among others, for the following sentences: R31-34, R41-46, R56-58, R69-70, R74-76, R96, R111-113, R117-118, R122, R125-136, R156, R178, R186-189, R198, R203-204, R230-231, R308-309, R332-334, R342-346, R348-351, R354-357, section 3.5 as a whole, R404-405, R412-413, R424-426, R427-431, R435, R458-462, ... (not further checked).
We sincerely apologize for any previous mistakes. We have carefully considered all your comments and made significant improvements to the paper. It has undergone a thorough revision and correction process by English-speaking experts to ensure accuracy, clarity, and language proficiency.
- The first chapter/section has the wrong lay-out (and is different from the rest of the manuscript).
Yes, we agree with the reviewer. We apologize again and now the whole paper has been changed and unified.
- References are missing in the first part of the paragraph starting at R35.
The references have been modified and positioned accordingly within the paragraph.R43-R48, R51.
- Table 1: Put the caption above the table. Why is the last row coloured? What does the * mean at Omicron?
We thank the reviewer for the appreciation because it seems this point was not clear. In fact, we have changed the table since the evolution of the Omicron variant has changed, for which it was coloured. The table has been significantly improved and thoroughly updated, resulting in enhanced clarity. R96-R113.
- The E in REASSURED is not indicated where it comes from.
The E is for REal connectivity, now it has been written in capital letters in the text. R159.
- R145: What about the position of the FDA?
We agree with the reviewer's perspective that incorporating the FDA's stance is crucial. As a result, we have now included the corresponding paragraph (R150-R154).
- Section 2: Why are only LFAs discussed here? Indicate clearly the boundaries of this review; what will be and what will not be discussed?
The purpose of this review is to elucidate the various diagnostic systems for COVID-19 based on nucleic acids. As our focus is primarily on nucleic acid amplification tests (NAATs), we have not dedicated a specific section to lateral flow assays (LFAs) since LFA relies on an immunochromatographic assay, which differs from NAATs. Nevertheless, we have included a brief explanation of LFA in the section titled 'Importance of Diagnosis' under the subsection 'Point-of-Care Tests (POCTs)'. We have done this for three reasons: i) LFA is one of the most commonly used molecular detection methods, ii) it is widely recognized as a point-of-care test worldwide, and iii) its mechanism shares similarities with other molecular diagnostic systems that utilize visual readouts. However, we have explicitly clarified the scope of our review in the concluding sentence of the paragraph, removing any previous ambiguity (R195-R202)."
- R179-180 and section 3.1: Give more details on the complete analysis process. Nucleic acid analysis is not only amplification, but also involves quite some sample work-up steps. Those steps usually involve quite some manual labour. So, also emphasize a bit more on what is meant with automatization.
We agree with the referee's comments. The sample preparation and the advantages/disadvantages of the automatization process of RT-qPCR have been included in the paper. R209-R211; R222-R224.
- R194: Emphasize more on the statement that the Ct-value might not be related to the viral load.
Thank you for the observation. We would like to emphasize that Ct-value is a measure of the amount of nucleic acid present in the sample and should not be interpreted as an indication of infectious viral particles. Previous steps of RNA isolation and reverse transcription could affect the quantification (Ct-values) R226.
- A ')' is missing somewhere around R220.
Thank you, now, it has been completed in the main text.
- The Xpert Xpress is not the only POCT with authorization from the FDA.
Thank you for the comment. We apologize if the paragraph implied that only Xpert was authorized. We have revised and rephrased the statement to ensure clarity and inclusivity. In addition, more examples have been included. R255-R260.
- There are no 'the' inhibitors (R234)
Thanks, the text has been changed. R267
- R257: this should be reverse instead of retro
Thanks, the text has been changed. R306
- LAMP can also be carried out with four primers.
Thanks, the text has been changed. R307-308
- What is meant with 'half complementarity' (R259)?
The original statement intended to convey that the sequence of external primers does not entirely bind to the target sequence, creating a new sequence with auto-complementarity that exposes a new binding site for the remaining primers (the loop), thus initiating the cascade of amplification. However, the statement has been revised to eliminate any potential confusion. R308-R310.
- R266: Although multiplexing is possible with LAMP, it is very challenging!
We agree with the referee and we have highlighted this question. We work with RPA, LAMP and multiplex, and indeed they are challenging and complex techniques with a promising future. R318-R320.
- From page 9 the lay-out is again incorrect.
Thanks, the text has been changed. Section 3.4.
- There is no figure 5 about RPA. Performance is not expressed in amount of minutes. What is meant with scanning the DNA? Explain better statements such as 'simpler and more sensitive'. The use of lateral flow is not restricted to RPA only. Which SARS-CoV-2 test uses RPA? Please emphasize more on these things.
Thanks for the comment. We apologize for the incorrect referencing of the figure in the text. We have now placed it in the correct position. We have changed the confused terms as "time of amplification", "scanning the DNA by recognizing the DNA target", for some simpler ones like "less quantity of enzymes involved" or "more sensitive by the amount of nucleic acid able to be detected". We also apologize from the text emerges LFA is only to RPA, we have changed the sentence to avoid confusion and include SARS-CoV-2 approaches that include RPA. R320-R340; R342-R350; R358-364
- R311 mentiones type I, II and III and then only something is said about a type V, which is very confusing.
We agree with the referee. We have carefully revised the text and provided a comprehensive explanation of the classification and the types of diagnoses that are particularly relevant. R378-382
- Lateral flow assay (LFA) is the correct term (R335).
Thanks, the text has been changed. R406
- MPS or NGS are more commonly used terms than HTS (section 3.5).
Thanks, it has been changed in the manuscript. section 3.5.
- Figure 2: What does minimum pretreatment enhold? Give more details. Time-to-result is a more common term. Be consistent with capitals. Not necessary.
We have provided more detailed explanations for each minimum pretreatment step, ensuring clarity and accuracy. Additionally, we have made appropriate changes to capitalization and revised the time to result for improved precision. Table 2. R637-640
- R547: I guess HIV is meant instead of VIH (R596)
Thanks. The paragraph has been reformulated, and the specified word has been removed
- The conclusion is missing an actual conclusion. What is the most promising technique?
Thank you for the comment, we strongly agree with the referee and conclusions have been revised to incorporate and clarify all the points we have discussed. R739-761.

Reviewer 2 Report
The review summarizes recent detection of SARS-CoV-2 based on Nucleic Acid Amplification tests and its integration in microfluidic platforms.The review can be considered for publication in "International Journal of Molecular Sciences" after revising the following questions. The comments are below.
1) In this review, the author summarizes the detection of Nucleic Acid Amplification tests. There is also a review which introduce SARS-CoV-2. I recommend introducing it in your background. Micromachines (Micromachines), 2022, 13(8), 1238. http://dx.doi.org/10.3390/mi13081238
2) Some incomplete words need to be modified like line 69:“VOCs have been defined by World Health Organization 68 (WHO) as variants that have to be observed due to higher transmissibility o detrimental change in COVID-19 epidemiology…”. Please verify the “o”.
3) Extra parentheses should be revised like line 218: “(different laboratories or patient care settings certified according to Clinical Laboratory Improvements Amendments of 1988 (CLIA) classifying…”
4) Some mistakes of grammar need to be modified like line 266: “It is possible multiplexing the 266 reaction or couple it with sequencing”. Please make sure “couple” or “coupling”.
5) A false reference from figure is at line 277: “The reaction is represented in Figure 5”
6) At line 356, the chip is presented abruptly and there is not chip in Figure 5. I recommend using Figure with chip to show CRISPR-based assays.
7) There is no Figure about microfluidic device or nanomedicine from part 4. I suggest introducing some figures about microfluidic device or nanomedicine. There are some suggested figure in two papers.
Biosensors and Bioelectronics (Biosens. Bioelectron.), 2022, 212, 114429. http://dx.doi.org/10.1016/j.bios.2022.114429
Frontiers in Chemistry (Front. Chem.), 2021, 9, 688442. http://dx.doi.org/10.3389/fchem.2021.688442
8) The logic is not suitable at line 545. The author firstly emphasize other tests have also been developed worldwide to help fast diagnosis, but the content after that is different from the first sentence. I suggest using other sentences to dominate the next sentences.
9) The clarity of Figures from the review should be improved.
Author Response
Comments and Suggestions for Authors
REFEREE #2
The review summarizes recent detection of SARS-CoV 2 based on Nucleic Acid Amplification tests and its integration in microfluidic platforms. The review can be considered for publication in "International Journal of Molecular Sciences" after revising the following questions. The comments are below.
1) In this review, the author summarizes the detection of Nucleic Acid Amplification tests. There is also a review which introduce SARS-CoV-2. I recommend introducing it in your background. Micromachines (Micromachines), 2022, 13(8), 1238. http://dx.doi.org/10.3390/mi13081238http://dx.doi.org/10.3390/mi13081238
Reply to referee. Thank you very much for your comment. We agree with the reviewer's assessment that this review is highly significant as it provides a concise and insightful introduction to SARs-CoV-2. Therefore, we have incorporated this review into our work to enhance its comprehensiveness. R197-R202.
2) Some incomplete words need to be modified like line 69: “VOCs have been defined by World Health Organization 68 (WHO) as variants that have to be observed due to higher transmissibility o detrimental change in COVID-19 epidemiology…”. Please verify the “o”.
Thanks for the comment, the text has been changed. R76-79
3) Extra parentheses should be revised like line 218: “(different laboratories or patient care settings certified according to Clinical Laboratory Improvements Amendments of 1988 (CLIA) classifying…”
Thanks for the comment, the text has been changed. R252.
4) Some mistakes of grammar need to be modified like line 266: “It is possible multiplexing the 266 reaction or couple it with sequencing”. Please make sure “couple” or “coupling”.
We sincerely apologize for any previous mistakes. We have carefully considered all your comments and made significant improvements to the paper. It has undergone a thorough revision and correction process by English-speaking experts to ensure accuracy, clarity, and language proficiency. R318-320.
5) A false reference from figure is at line 277: “The reaction is represented in Figure 5”
Thanks for the comment. We apologize since there was a mistake in the number of this figure. R344
6) At line 356, the chip is presented abruptly and there is not chip in Figure 5. I recommend using Figure with chip to show CRISPR based assays.
We agree with the reviewer's suggestion and Figure 5 now shows the chip described in the text. Figure 4. R433
7) There is no Figure about microfluidic device or nanomedicine from part 4. I suggest introducing some figures about microfluidic device or nanomedicine. There are some suggested figure in two papers. Biosensors and Bioelectronics (Biosens. Bioelectron.), 2022, 212, 114429. http://dx.doi.org/10.1016/j.bios.2022.114429 and Frontiers in Chemistry (Front. Chem.), 2021, 9, 688442. http://dx.doi.org/10.3389/fchem.2021.688442
We strongly agree with the reviewer's suggestion. We have included a new figure (Figure 5) about the main nanomedicine-based molecular diagnostic systems for SARS-CoV-2 currently in use: microfluidics, plasmonic sensing and raman spectroscopy. R559
8) The logic is not suitable at line 545. The author firstly emphasize other tests have also been developed worldwide to help fast diagnosis, but the content after that is different from the first sentence. I suggest using other sentences to dominate the next sentences.
We agree with this comment.The paragraph has been reformulated, and thi sentence has been removed. R662-670.
9) The clarity of Figures from the review should be improved
We strongly agree with the reviewer's suggestion. All figures have been replaced with clearer and higher-quality versions to enhance their visual clarity.

Round 2
Reviewer 1 Report
The authors did a great job on editing the English of the manuscript. However, there are still some issues that need to be addressed upon publication.
1. In the abstract the topics of the first two section are not mentioned.
2. The article starts with a too extensive introduction and importance of diagnosis (section 2). These topics are not introduce properly and contain too much details. A bit of background information is fine, but these sections discuss issues that are not directly related to NAAT tests.
3. In general an overview/outline of the review is missing. Usually this is given at the end of the (short) introduction.
4. Figure 1: The last part of the caption should be put in the text, since it is not directly related to what can be seen in the figure.
5. While used correctly is it helpful to explain the difference between SARS-CoV-2 (the virus/analyte) and COVID-19 (the illness).
6. In many countries RT-PCR is not used in a quantitative manner. There are even articles that claim that the Ct-value is not always related to the amount of virus particles (it is even mentioned in section 3.1). Usually for SARS-CoV-2 RT-PCR is only used as a qualitative method (i.e. only yes/no answer).
7. In section 2.1 there are a lot of details on the principle of LFA tests, while this is not the topic of this article.
8. The policy around the use of LFAs and NAAT tests differs from country to counrtry. Therefore, the statments made at the end of 2.1 are a bit bold. Also the rules around the use of LFAs could change along the pandemic. E.g. at first instance LFAs could only be used when you were symptomatic and later on that was not needed anymore. And in the beginning a confirmatory NAAT test was required after a postive LFA test and later on only LFA was enough. So, please rephrase this part.
9. Usually the content of a review is mentioned (early on in the manuscript), but not the objective of such a review. It is advised to remove the first sentence of this paragraph. And please rehprase the rest a bit and move it to the beginning of the manuscript providing a guidance for the reader on what to expect in the article.
10. In section 3.1 two steps are mentioned, but three steps are discussed (extraction, reverse transcription and amplification).
11. It seems that Figure 4 does not have a caption anymore.
12. The relation between NAAT test on one hand and the microfluidics/nanomedicine on the other hand is not explained enough. Again guidance for the reader is missing.
13. The level of details in table 2 differs from test to test quite a lot. In the beginning the extraction protocol is explained and later on only 'extraction' is mentioned. Sometimes the unit is written out and sometimes not. The table is quite chaotic and it would make the table more readable if for instance the type of tests are ordered/after each other.
I still do not see the knowledge gap mentioned by the authors. When I use a search engine and look for articles about SARS-CoV-2, NAATs, POC and/or microfluidics I find many reviews. Some examples:
* Tools and Techniques for Severe Acute Respiratory Syndrome Coronavirus 2 (SARS-CoV-2)/COVID-19 Detection - PMC (nih.gov)
* Advances in laboratory detection methods and technology application of SARS‐CoV‐2 - PMC (nih.gov)
* Laboratory Diagnosis of SARS-CoV-2 Pneumonia - PMC (nih.gov)
* FDA authorized molecular point-of-care SARS-CoV-2 tests: A critical review on principles, systems and clinical performances - PMC (nih.gov)
* An Update on Molecular Diagnostics for COVID-19 - PMC (nih.gov)
* A systematic review of the advancement on colorimetric nanobiosensors for SARS-CoV-2 detection - PMC (nih.gov)
Moreover, the authors aim to be not too technical, but the first two sections contain a lot of details and goes a lot into depth.
Overall, the structure of the manuscript is not very clear. Guidance for the reader is missing (e.g. an outline in the introduction). And as mentioned above is the newness of this review not clear enough.
Author Response
Second Round
REFEREE 1
The authors did a great job on editing the English of the manuscript.
We thank the referee. We have done a hard job thanks to your comments/suggestions.
However, there are still some issues that need to be addressed upon publication.
- In the abstract the topics of the first two section are not mentioned.
Reply to referee: Thank you very much for this comment. We have included that information in the revised version of the manuscript.
- The article starts with a too extensive introduction and importance of diagnosis (section 2). These topics are not introduced properly and contain too much details. A bit of background information is fine, but these sections discuss issues that are not directly related to NAAT tests.
We appreciate this comment, and as a result, the introduction and section 2 have been shortened and rephrased. Indeed, section 2 has been significantly condensed, with a focus on POCT and NAATs. Additionally, following the referee's recommendation and in order to ensure proper introduction of certain topics, we have included a brief overview of the article in the introduction (section 1).
- In general an overview/outline of the review is missing. Usually this is given at the end of the (short) introduction.
We agree with the referee's suggestion that an overview of the review is necessary for all papers. In the previous revision, a paragraph was inserted to fulfill this requirement. However, it appears that the clarity of that paragraph was insufficient. Therefore, in this new revised version of the article, we have included a clearer overview to ensure the reader's understanding of the paper's content (R107-114).
- Figure 1: The last part of the caption should be put in the text, since it is not directly related to what can be seen in the figure.
We agree with that, this sentence has now been included in the text (R55-R57).
- While used correctly is it helpful to explain the difference between SARS-CoV-2 (the virus/analyte) and COVID-19 (the illness).
A new sentence has now been included at the beginning of the article (R42-43).
- In many countries RT-PCR is not used in a quantitative manner. There are even articles that claim that the Ct-value is not always related to the amount of virus particles (it is even mentioned in section 3.1). Usually for SARS-CoV-2 RT-PCR is only used as a qualitative method (i.e. only yes/no answer).
Thank you very much for this comment. This explanation is now included in the revised version of the manuscript (R181-R183).
- In section 2.1 there are a lot of details on the principle of LFA tests, while this is not the topic of this article.
We agree and the text has been modified accordingly. In the revised manuscript LFA tests were only mentioned in one sentence to compare with NAATs (R133-R135) and another sentence to include the information you suggested in comment 8 (next comment) in the manuscript.
- The policy around the use of LFAs and NAAT tests differs from country to counrtry. Therefore, the statments made at the end of 2.1 are a bit bold. Also the rules around the use of LFAs could change along the pandemic. E.g. at first instance LFAs could only be used when you were symptomatic and later on that was not needed anymore. And in the beginning a confirmatory NAAT test was required after a postive LFA test and later on only LFA was enough. So, please rephrase this part.
Thank you for this appreciation, we took it into consideration and this information is now included in the manuscript (R135-R138).
- Usually the content of a review is mentioned (early on in the manuscript), but not the objective of such a review. It is advised to remove the first sentence of this paragraph. And please rehprase the rest a bit and move it to the beginning of the manuscript providing a guidance for the reader on what to expect in the article.
This paragraph has been rewritten again (R107-114).
- In section 3.1 two steps are mentioned, but three steps are discussed (extraction, reverse transcription and amplification).
We agree with the referee and it has been modified R163-R168.
- It seems that Figure 4 does not have a caption anymore.
We have checked again all the figures (also number 4) and we have verified that all of them have its caption.
- The relation between the NAAT test on one hand and the microfluidics/nanomedicine on the other hand is not explained enough. Again guidance for the reader is missing.
Thank you very much for this comment. We have included an explanation of that relationship in the manuscript. It was introduced in R145-R158 and then discussed on R452-R461.
- The level of details in table 2 differs from test to test quite a lot. In the beginning the extraction protocol is explained and later on only 'extraction' is mentioned. Sometimes the unit is written out and sometimes not. The table is quite chaotic and it would make the table more readable if for instance the type of tests are ordered/after each other.
Thank you very much for this comment. We agree and we have changed the table according to the referee´s suggestion. Now it is ordered by the type of test and information has been revised and completed (R615-617).
I still do not see the knowledge gap mentioned by the authors. When I use a search engine and look for articles about SARS-CoV-2, NAATs, POC and/or microfluidics I find many reviews. Some examples:
Authors: Thank you for the references included in this comment. They were very useful to enrich our review. We have read all these articles and we can identify several differences with the review presented here. In order to recognize the knowledge gap, we have prepared a concise summary highlighting the variations between each article and our manuscript:
* Tools and Techniques for Severe Acute Respiratory Syndrome Coronavirus 2 (SARS-CoV-2)/COVID-19 Detection - PMC (nih.gov)
It talks about different diagnosis methods (RT-PCR, RPA- TMA, CRISPR-Cas, ELISA, CLIA…) but it didn’t address the nanomedicine, microfluidic or POCT approach.
* Advances in laboratory detection methods and technology application of SARS‐CoV‐2 - PMC (nih.gov)
The mentioned article discusses various diagnostic methods but does not cover the topics of nanomedicine or microfluidics.
* Laboratory Diagnosis of SARS-CoV-2 Pneumonia - PMC (nih.gov)
The article discusses several diagnostic methods such as RT-PCR, isothermal techniques like RT-LAMP/NEAR, RT-PCR/CRISPR, and other tests. While it touches briefly on the topic of POCT, it does not specifically address the nanomedicine or microfluidic approach.
* FDA authorized molecular point-of-care SARS-CoV-2 tests: A critical review on principles, systems and clinical performances - PMC (nih.gov)
This article primarily focuses on POCT, but similar to the previous case, it does not provide any information regarding the nanomedicine or microfluidic approach.
* An Update on Molecular Diagnostics for COVID-19 - PMC (nih.gov)
This article has only a table about Rt-qPCR, NASBA, LAMP, CRISPR-based, antigens. POCT, but again it didn’t address the nanomedicine or microfluidic approach.
* A systematic review of the advancement on colorimetric nanobiosensors for SARS-CoV-2 detection - PMC (nih.gov)
The article discusses nanomedicine, specifically silver nanoparticles (AgNPs), magnetic nanoparticles (MNPs), carbon-based nanostructures, and nanohybrid core-shell structures. However, it is important to note that this particular article does not focus on NAATs or their adaptations in conjunction with microfluidics and nanomedicine.
Moreover, the authors aim to be not too technical, but the first two sections contain a lot of details and goes a lot into depth.
We hope this is now sorted out since most of the technical parts of the first two sections were erased from the manuscript.
Overall, the structure of the manuscript is not very clear. Guidance for the reader is missing (e.g. an outline in the introduction). And as mentioned above is the newness of this review not clear enough.
Thank you for your review. We have addressed all the referee´s comments and suggestions to improve the manuscript and we hope it is now suitable for publication.

Reviewer 2 Report
The authors seem to have worked hard on the revision. I have no more comments.
Author Response
We thank the referee. We have done a hard job thanks to your comments/suggestions.
Round 3
Reviewer 1 Report
R286: no (instead of not)
Figure 5: what is the source of this picture?
Author Response
REFEREE 1
R286: no (instead of not)
We apologize to the referee for this mistake. It has now been corrected in the manuscript.
Figure 5: what is the source of this picture?
We thank the referee for the question and the opportunity to explain it. Referee 2 has emphasized the significance of including a figure depicting the principal nanomedicine-based molecular diagnostic systems currently employed for SARS-CoV-2, such us microfluidics, plasmonic sensing, and Raman spectroscopy. It is important to note that this figure is an original creation and does not replicate any existing publication. We generated the figure using Biorender software, as explained in the Acknowledgments section.
